# Molecular Alterations in Gastric Preneoplastic Lesions and Early Gastric Cancer

**DOI:** 10.3390/ijms22136652

**Published:** 2021-06-22

**Authors:** Serena Battista, Maria Raffaella Ambrosio, Francesco Limarzi, Graziana Gallo, Luca Saragoni

**Affiliations:** 1Pathology Department, “S. Maria della Misericordia Hospital”, Friuli-Venezia Giulia, 33100 Udine, Italy; 2Pathology Department, “Az. USL, Tuscany North-West”, Tuscany, 56121 Pisa, Italy; maradot@libero.it; 3Pathology Department, “G.B. Morgagni-L. Pierantoni Hospital”, Emilia-Romagna, 47121 Forlì, Italy; francesco.limarzi@auslromagna.it (F.L.); luca.saragoni@auslromagna.it (L.S.); 4Pathology Department, “M. Bufalini Hospital”, Emilia Romagna, 47521 Cesena, Italy; graziana.gallo@auslromagna.it

**Keywords:** gastric preinvasive neoplasia, early gastric cancer, signet ring cell gastric carcinoma, mucins expression, molecular markers

## Abstract

Prognosis of gastric cancer is dramatically improved by early diagnosis. Correa’s cascade correlates the expression of some molecular markers with the progression of preneoplastic lesions toward carcinoma. This article reviews the diagnostic and prognostic values of molecular markers in complete (*MUC2*) and incomplete (*MUC2*, *MUC5AC*, and *MUC6*) intestinal metaplasia, gastric dysplasia/intra-epithelial neoplasia, and early gastric cancer. In particular, considering preinvasive neoplasia and early gastric cancer, some studies have demonstrated a correlation between molecular alterations and prognosis, for example, mucins phenotype in gastric dysplasia, and GATA6, TP53 mutation/LOH and *MUC6* in early gastric cancer. Moreover, this review considers novelties from the literature regarding the (immuno)histochemical characterization of diffuse-type/signet ring cell gastric cancer, with particular attention to clinical outcomes of patients. The aim of this review is the evaluation of the state of the art regarding suitable biomarkers used in the pre-surgical phase, which can distinguish patients with different prognoses and help decide the best therapeutic strategy.

## 1. Introduction

Worldwide, gastric cancer ranks fifth in incidence and fourth in mortality, registering one million new cases and an estimated 769,000 deaths in 2020 [1]. There are two topographical subtypes, the cardia (upper stomach) and non-cardia (lower stomach), which have different risk, carcinogenesis, and epidemiologic factors. Incidence and mortality rates of non-cardia cancer have been decreasing over the last fifty years due to prevention (e.g., *Helicobacter pylori* infection and storage of foods), although a significant increase in the incidence of stomach cancer among young adults (aged < 50 years) was reported [2].

Both gastric precancerous lesions and gastric cancer are associated with a spectrum of genetic and epigenetic abnormalities [3,4,5]. These include genetic instabilities and abnormalities in oncogenes, tumor suppressor genes, growth factors, receptor tyrosine kinases, DNA repair genes, matrix degradation enzymes, cell-cycle regulators, and cell adhesion molecules. Somatic mutations may occur in proliferative and preinvasive lesions (e.g., mutations of CTNNB1 in fundic gland polyps, GNAS in pyloric gland adenomas, and TP53 in high-grade dysplasia and adenocarcinoma) [6,7,8,9,10].

According to Lauren’s classification, gastric carcinoma is classified as intestinal-type and diffuse-type [11]. Intestinal-type cancers show biological and epidemiological features that are different from those of diffuse-type cancers [12]. These data suggest that tumors displaying different biological aspects are associated with a distinct clinical behavior.

Recently, new molecular classifications of gastric cancer were introduced. In 2014, based on key DNA defects and molecular abnormalities, The Cancer Genome Atlas Consortium (TCGA) divided gastric cancers into Epstein-Barr virus (EBV)-positive, microsatellite instability (MSI), gene stable (GS), and chromosome instability (CIN) types [13]. TCGA typing is based on European and U.S. populations; however, the clinical characteristics of TCGA typing in the Asian population and its association with clinical parameters and prognosis remain unclear.

To prevent the development of gastric cancer, the diagnosis of precancerous lesions is crucial and should be encouraged, especially in countries where organized screening programs do not exist, and in areas with a high incidence of gastric cancer. In Japan, where organized screening programs are still active, more than 50% of gastric cancers are diagnosed in the early phase. Patients affected by early gastric cancers have a good prognosis, with overall survival rates close to 100% after 5 years of follow up and can be treated with radical endoscopic resection (ESD) [14].

Another important aspect of early diagnosis and prevention is hereditary diffuse gastric cancer (HDGC), characterized by the prevalence of diffuse gastric cancer and lobular breast cancer. It is largely caused by inactivating germline mutations in the tumor suppressor gene CDH1, although pathogenic variants in CTNNA1 occur in a minority of families with HDGC.

Recent clinical practice guidelines for HDGC from the International Gastric Cancer Linkage Consortium (IGCLC), which recognizes the emerging evidence of variability in gastric cancer risk between families with HDGC, focus on the growing capability of endoscopic and histological surveillance in HDGC and increased experience of managing the long-term sequelae of total gastrectomy in young patients. Prophylactic total gastrectomy remains the recommended option for gastric cancer risk management in pathogenic CDH1 variant carriers. However, there is increasing confidence from the IGCLC that endoscopic surveillance in expert centers can be safely offered to patients who wish to postpone surgery or to those whose risk of developing gastric cancer is not well-defined [15]. The precancerous stages of intestinal-type gastric cancer represent a complex process, part of which results in a transformation of the normal mucosa to an intestinal metaplastic mucosa through a series of lesions forming a continuum. This sequence of events may last for several years and is designated as Correa’s cascade of multistep gastric carcinogenesis [16]. According to this scheme, the morphological changes observed fall into three categories: chronic inflammation of the gastric mucosa (chronic gastritis), mucosal loss of appropriate gastric glands (atrophy), and substitution of gastric epithelium with intestinal epithelium (intestinal metaplasia).

Through successive mutations, the gastric epithelial cells disappear and are replaced by cells with an intestinal phenotype, which, over time, gain autonomy, favoring the development of dysplastic changes (intraepithelial neoplasia) and carcinoma.

Correa’s cascade accounts for the carcinogenesis of intestinal-type gastric cancer. In this regard, the OLGA system provides a basis for predicting gastric cancer risks associated with atrophic gastritis and intestinal metaplasia, as well as guiding clinical surveillance [16,17].

Intestinal-type adenocarcinomas are subclassified into three groups: foveolar, intestinal, and combined. These three groups have different expressions of mucins, have distinct cellular mucin phenotypes resulting from different genetic alterations, and are clinically relevant. Similarly, recent studies reported different patterns of mucins expression in preneoplastic lesions, suggesting biological specific pathways. Complete intestinal metaplasia shows goblet cells, absorptive enterocytes with a luminal brush border, and intestinal mucin (*MUC2*) expression. In contrast, incomplete intestinal metaplasia displays goblet cells, absorptive cells without a brush border, and co-expression of intestinal (*MUC2*) and gastric (*MUC5AC*, *MUC6*) mucins [18].

In this review, we summarize the most recent knowledge of mucins expression in preneoplastic and neoplastic lesions of the stomach, with emphasis on their precise clinicopathologic and prognostic role.

Unlike the intestinal-type, the diffuse-type gastric cancer, according to Lauren, had never been studied for mucins expression before the recent paper by the European Chapter of the International Gastric Cancer Association [19]. In this study, researchers found that mucins do not help in distinguishing signet ring cell from non-signet-ring cell gastric carcinoma. However, mucin stains expression helps identify different outcomes. Furthermore, outcomes and mucins expression seem to differ between Caucasian and Asian patients.

## 2. Mucins Expression in Normal Gastric Mucosa

Mucins are glycoproteins of gastric mucus that protect gastric mucosa and are normally expressed in non-neoplastic gastric tissue [20]. Their overexpression and decrease in the immune response are associated with neoplastic transformation. Their levels correlate with the histological origin of the tumor (e.g., *MUC5AC* is secreted by antrum and body foveolar cells, and *MUC6* by pyloric glands) [21]. Their expression may be associated with clinical outcomes [19,21]. To date, twelve mucin genes that code for the protein part of mucins (apomucins) have been identified: *MUC1*, *MUC2*, *MUC3*, *MUC4*, *MUC5AC*, *MUC5B*, *MUC6*, *MUC7*, *MUC8*, *MUC9*, *MUC11*, and *MUC12*. In particular, *MUC2*, *MUC5AC*, *MUC5B*, and *MUC6* genes have been mapped in a cluster on chromosome 11p15.5; and *MUC3*, *MUC11*, and *MUC12* have been mapped in another cluster on chromosome 7q22.

In situ hybridization and immunohistochemical studies have shown that these mucins are differently expressed in epithelia with cell type specificity.

Normal gastric mucosa produces mainly neutral mucins, except for the mucus-secreting cells of the neck glands that secrete acid mucins. The normal gastric mucosa shows cell types specifically expressing *MUC1*, *MUC5AC*, and *MUC6*, with the first two mucins found in the superficial foveolar epithelium (*MUC5AC* is also highly expressed in mucous neck cells of the antrum), whereas *MUC6* is expressed in the mucous neck cells of the body and deeper glands of the antrum. Normal gastric mucosa does not express *MUC2*.

Changes in the expression levels and glycosylation patterns of mucins were associated with several diseases, including preneoplastic lesions and gastric carcinoma [22]. These observations suggest that mucins alterations can be regarded as molecular markers of malignant transformation to gastric mucosa. Furthermore, some studies strongly suggested *MUC5AC* as a putative *H. pylori* receptor.

## 3. Gastric Mucosal Atrophy

In the gastric mucosa, longstanding inflammation can trigger phenotypic changes: fibrosis of the lamina propria (replacing glandular loss) and/or metaplastic transformation of the native glands (mucosecreting or oxyntic). These changes result in mucosal atrophy [16,17].

The definition of mucosal atrophy encompasses both the disappearance and the metaplastic (intestinal and/or pseudopyloric) transformation of the native glands. Mucosal atrophy results in functional changes affecting the production of acid and the secretion of pepsinogen and gastrin. Gastric mucosal atrophy is the cancerization field of non-syndromic gastric cancers [16].

However, a direct role of pseudopyloric metaplasia in the histogenesis of gastric cancer is more widely debated [23,24]. Pseudopyloric metaplasia features the same morphology and TFF2 expression seen in pyloric-type glands [23]. Pseudopyloric metaplasia may also undergo further changes toward intestinalization.

## 4. Intestinal Metaplasia

Intestinal metaplasia (IM) represents a substitution of the gastric epithelium with intestinal epithelium. There are two types of intestinal metaplasia: the complete type, or type I, with absorptive cells, Paneth cells, and goblet cells secreting sialomucins (the small intestine phenotype); and the incomplete type, or types II and III, with columnar and goblet cells secreting neutral and acid sialomucins in type II and sulfomucins in type III [18,23]. Several studies demonstrated that type III intestinal metaplasia has a greater risk of malignant transformation than types I and II intestinal metaplasia. Several studies have highlighted underexpression of *MUC1*, *MUC5AC*, and *MUC6*, both in goblet and columnar cells. Moreover, the de novo expression of intestinal *MUC2*, with >75% of positive cells with a diffuse cytoplasmic pattern in goblet cells, has been demonstrated. Specifically, in type I, or complete IM, the expression of *MUC2* intestinal mucin and decreased/absent expression of *MUC1*, *MUC5AC*, and *MUC6* are observed. Instead, in type II/III, or incomplete IM, there is coexpression of *MUC2* and the other mucins normally expressed in the stomach [18,23,24,25,26]. The most extensively researched homeobox genes in stomach cancer are CDX2 and CDX1. These genes are closely involved in the development of the intestinal metaplasia of gastric mucosa. CDX1 transgenic mouse model showed that the replacement of gastric mucosa with IM involves all four epithelial cell types (absorptive enterocytes, goblet, enteroendocrine, and Paneth cells), whereas only pseudopyloric gland metaplasia was observed in the CDX2 transgenic mouse. In the human stomach, ectopic expressions of CDX1 and CDX2 have been demonstrated frequently in IM. However, only CDX2 has been observed as an independent factor for the development of intestinal-type gastric adenocarcinoma. Moreover, it was shown that the expression of CDX2 in gastric cancer is mainly due to promoter hypermethylation. This finding suggests that aberrant downregulation of CDX2 might promote gastric carcinogenesis. In addition, the expression of CDX2 protein has the highest values in complete IM, followed by incomplete IM and gastric cancer, which has the lowest values. Finally, using RT-PCR, both CDX1 and CDX2 mRNAs have been detected in mild and severe IM types. However, none of these mRNAs has been identified in normal gastric mucosa without IM, suggesting their expression may contribute to the intestinal phenotype. The high levels of these mRNAs in IM mucosa, associated with chronic atrophic gastritis, suggest an association with this phenotypic shift in the gastric mucosa [26,27].

## 5. Gastric Dysplasia

The latest WHO classification [28] defines gastric dysplasia/intra-epithelial neoplasia as an unequivocally neoplastic modification of the stomach epithelium without evidence of stromal invasion. The two broad categories of dysplasia in the stomach are intestinal and foveolar (gastric) types; however, some cases are mixed. Other subtypes include gastric pit/crypt dysplasia and serrated dysplasia. Dysplasia occurs de novo or within pre-existing benign lesions such as a fundic gland polyp or a hyperplastic polyp. Dysplasia, regardless of the morphological subtype, is defined either as low or high grade, according to the WHO classification, which considers nucleus atypia, mitotic activity, cytoplasmic differentiation, and architectural disarray as diagnostic criteria.

Intestinal dysplasia consists of tubular, tubulo-villous, or villous lesions lined by tall columnar cells. Terminal differentiation toward absorptive-type cells, goblet cells, endocrine cells, or Paneth cells is common. Accordingly, intestinal-type dysplasia is immunoreactive for *MUC2*, CD10, and CDX-2, and resembles colonic dysplasia [28].

Foveolar dysplasia has tubulo-villous fronds and serrated aspects with the presence of apical neutral mucins and immunoreactivity for *MUC5AC* and *MUC6* (at the bottom of the glands); *MUC2*-positive goblet cells are usually absent. Gastric pit/crypt dysplasia shows dysplasia at the basal portion of gastric pits and mucins expression recapitulates that observed in dysplasia of the intestinal-type. Serrated dysplasia is characterized by a micropapillary pattern with pit region dysplasia and *MUC5AC* expression.

Generally, in gastric dysplasia, the expression of *MUC5AC* and *MUC6* is more easily detected compared to intestinal metaplasia, where *MUC2* is typically expressed. *MUC5AC* and *MUC6* display a diffuse cytoplasmic expression pattern in columnar cells, whereas *MUC2* expression shows a diffuse cytoplasmic pattern in goblets cells.

### 5.1. Gastric Adenoma

Gastric adenoma is considered a preneoplastic lesion. There are different types: plain, villous, pedunculated, or depressed. Depending on differentiation, two types are distinguished: intestinal, with goblet cells and/or Paneth cells (intestinal-type); or dysplastic foveolar epithelium (foveolar type). The intestinal-type is associated with a mucosa that has intestinal metaplasia and gastric atrophy, with about a 40% occurrence of high-grade dysplasia. The foveolar type is found in normal epithelium, non-atrophic mucosa and is not associated with high-grade dysplasia or synchronous adenocarcinoma.

With immunohistochemical staining, we can distinguish the different types. Intestinal-type gastric adenomas show *MUC2* and/or CDX2 expression, whereas the foveolar type is characterized by strong/diffuse *MUC5AC* positivity, weak/focal *MUC6* immunostaining, and an absence of *MUC2* or CDX2 expression [28,29].

### 5.2. Gastric Pyloric Gland Adenomas

Pyloric gland adenoma is a gastric epithelial polyp consisting of neoplastic pyloric type glands lined by cuboidal/low columnar epithelia. High-grade dysplasia is observed in 40–50% of cases. Consistent with the pyloric type phenotype, pyloric gland adenomas variably express *MUC6*. *MUC5AC* positivity may involve the whole lesion with no restriction to surface epithelium. Low-grade lesions mostly feature diffuse *MUC6* expression, with a surface coating of *MUC5AC*. About 10% of pyloric gland adenomas display focal expression of *MUC2* and/or CDX2 [28,29].

### 5.3. Oxyntic Adenomas

Oxyntic gland adenoma is a benign epithelial neoplasm composed of columnar cells with differentiation to chief cells, parietal cells, or both, characterized by a high rate of progression to adenocarcinoma. The predominant cell type is represented by immature chief cells, which are positive for *MUC6* [28,29].

## 6. Early Gastric Cancer

The term early gastric cancer (EGC) and its definition as a carcinoma limited to the mucosa and/or submucosa, regardless of lymph-node status, was first proposed in 1971 and then included in the guidelines of the Japanese Gastric Cancer Association [30].

This definition has been criticized, especially with the rise in endoscopic treatment for early lesions. Many studies were conducted, focusing on parameters that can be associated with adverse prognosis, treatment failure, or lymph node metastases in ECG, without reaching a definitive consensus. The main problem originates from the lack of clear criteria distinguishing ECG with excellent prognosis (>98% 5-year survival) from ECG with higher incidence of lymph node metastases and worse prognosis (70% 5-year survival).

The presence of lymph node metastases at diagnosis is due to predictive parameters, including growth patterns of infiltration of the submucosa, according to Kodama’s classification (Table 1). Kodama’s PEN A type growth patterns are independent negative prognostic factors, identifying tumors with clinical behavior similar to that of advanced cancers, as we have published in a few previous studies [31,32,33,34].

Many parameters can be implicated in the different prognoses of EGC subtypes, but they have not been thoroughly explored. In the 1980s, Inokuchi demonstrated the correlation between a different cell nuclear DNA distribution pattern and a malignancy in PEN A that is characterized by aneuploid and a high-ploidy DNA range, such as carcinoma [35]. In recent studies, the aggressiveness of EGC correlated with a tumor microenvironment and genomic features (e.g., *MUC1* expression and gastric carcinoma) [36,37,38].

In 1999, Egashira et al. [39] showed that the differentiated (DA) minute (<5 mm) adenocarcinomas with gastric phenotype differ morphologically and histogenetically from DA with intestinal phenotype, as the former lack intestinal metaplasia in the surrounding non-neoplastic mucosa. However, they demonstrated that as the tumor with gastric phenotype grows, intestinal metaplasia progresses, intestinal-type phenotypic expression appears, and then DA with gastric phenotype changes into DA with gastric-intestinal or intestinal phenotype.

In another study, the histologic conversion from differentiated type carcinoma (DC) to undifferentiated type carcinoma (UDC) seemed to occur mainly in gastrointestinal mucin phenotype (GIM-type) and gastric mucin phenotype (GM-type) tumors, depending on the size of the lesion. In other words, with an increase in tumor size, small DCs with GIM and GM phenotypes might change histologically into UDCs. Instead, DCs with intestinal mucin (IM) phenotypic expression rarely show histologic conversion [35].

Regarding molecular alterations, Tsukashita, in 2001 [36], evaluated the histogenesis of gastric adenocarcinoma by *MUC* gene expression in eighty intramucosal gland-forming tumors. These tumors were categorized in three groups (according to the Vienna classification): group A (low-grade adenoma/dysplasia), group B (high-grade adenoma/dysplasia), and group C (intramucosal carcinoma). There were different expressions of the *MUC* genes: in group A, the lesions expressed intestinal markers and had a stable intestinal phenotype; whereas more than 50% of group B and C tumors expressed gastric markers, were unstable, and should have been considered de novo carcinomas.

Types II and III intestinal metaplasia tend to show mixed differentiation, including gastric- and intestinal-type cells, often expressing *MUC2*, *MUC5AC*, and *MUC6* at the same time. Gastric mucosa surrounding minute adenocarcinomas also appears to be intestinalized and the metaplasia is of the incomplete type. The results of this study, in which *MUC* gene expression in incomplete form of intestinal metaplasia was similar to that in group B and C tumors, also support this hypothesis.

Moreover, despite the small number of cases, a recent article demonstrated the possibility to differentiate between Pen A and Pen B EGCs using an immunohistochemical stain for mucin *MUC6* (more expressed in Pen A tumors) and analyzing the copy number of GATA6 (more frequently amplified in Pen B types), introducing the possibility of distinguishing these two types of lesions in small pre-operatory biopsies [38]. In this study, Molinari et al. analyzed 33 Pen A, 34 Pen B, and 20 T3N0 tumors (control group) and performed immunohistochemistry for mucins, copy number variation analysis of a gene panel, microsatellite instability (MSI), TP53 mutation, and loss of heterozygosity (LOH) analyses. The results showed that the Pen A subgroup was significantly characterized by *MUC6* overexpression (*p* = 0.021). Otherwise, the Pen B type was significantly associated with the amplification of the GATA6 gene (*p* = 0.002). A higher percentage of MSI tumors was observed in the T3N0 control group (*p* = 0.002), but no significant differences between the two EGC subtypes were found. Finally, the TP53 gene analysis showed that 32.8% of Pen tumors had a mutation in exons 5–8 and 50.0% presented LOH. The co-occurrence of the TP53 mutation and LOH mainly characterized Pen A tumors (*p* = 0.022).

The result is that clinic-pathologic parameters, microsatellite status, and frequency of TP53 mutations do not seem to distinguish Pen subgroups. Conversely, the amplification of GATA6 is associated with Pen B tumors, and the overexpression of *MUC6* and the TP53mut/LOH significantly characterizes Pen A lesions (Figure 1, Figure 2 and Figure 3).

All the photos are from the same lesion, were published with permission from Francesco Limarzi, and were not previously published in another journal.

Considering the signet ring cell carcinoma (SRC-GC) that falls into the group of diffuse-type gastric cancer and shows a better prognosis in its early phase compared to poorly cohesive intramucosal cancers without signet ring cells, 12 studies on histochemical mucins expression were published between 1977 and 2013, according to Lauren. The studies of Kubota, Akamatsu, and Tatematsu distinguished SRC-GC into several subtypes. Kubota et al. [40] used Alcian Blue (AB), Alcian Blue-PAS (AB-PAS), and LNAase stain to classify 64 SRC-GC as type A (immature: PAS weak positive, AB negative, LNAase positive, small cell size, and high nuclear/cytoplasmic ratio), type B (intermediate: stronger PAS positivity, AB-negative or -weak-positive, LNAase-positive, and smaller nuclear/cytoplasmic ratio), or type C (mature: PAS-strong-positive, AB -positive, and eccentric nucleus).

Akamatsu et al. [41] used five stains, AB-PAS, HID-AB, GOS, PA-SB-PH-PAS, and PCS to distinguish 31 SRC-GC into six subtypes: surface mucous cell-type, mucous neck cell-pyloric cell-type, goblet cell (small intestine)-type, goblet cell (large intestine)-type, microcyst-type, and unclassified.

Tatematsu et al. [42,43] used PCS, GOS, and sialidase GOS with immunohistochemical stains for pepsinogen I and II to distinguish 127 SRC-GC as gastric phenotype, intestinal phenotype, or mixed gastrointestinal phenotype, with the gastric phenotype resulting in the most prevalent subtype.

These subclassifications of SRC-GC have not been validated in other studies.

In order to better define signet ring cell carcinoma and its prognosis compared to poorly cohesive gastric cancer without SRCs, Kerckhoffs et al. [19], in an article which, to the best of our knowledge, is the largest study where all cancers were reclassified in a standardized manner according to WHO classification, compares Asian and Caucasian patients for the first time according to mucins expression, considering the relationship between mucins expression and patient outcome. The article shows no immunohistochemical mucins stain unique to SRC-GC. However, the mucins expression may be related to the quantity of SRCs within a given tumor, as the authors noticed a more frequent expression of mucins in poorly cohesive gastric cancer/diffuse gastric cancer containing > 10% SRCs. In their series, there are poorly cohesive cancers with ≥10% SRCs expressed more frequently: *MUC2*, *MUC5AC*, and ABPAS (*p* < 0.001, *p* = 0.004, and *p* < 0.001, respectively). From a prognostic point of view, patients with *MUC2* positive SRC-GC or SRC-GC with (gastro)intestinal phenotype have the poorest outcome. Moreover, Caucasians with AB-positive GC or combined ABPAS-*MUC2*-positive and *MUC5AC*-negative have the poorest outcome (all *p* = 0.002), whereas this association is not seen in Asian patients.

## 7. Conclusions

In this review, we summarized the most recent knowledge on mucins phenotype in preneoplastic and neoplastic lesions of the stomach, with particular emphasis on their clinico-pathologic correlations and prognostic role.

Regarding preneoplastic lesions and early cancers, many studies have demonstrated the important prognostic role played by mucins phenotype.

## Figures and Tables

**Figure 1 ijms-22-06652-f001:**
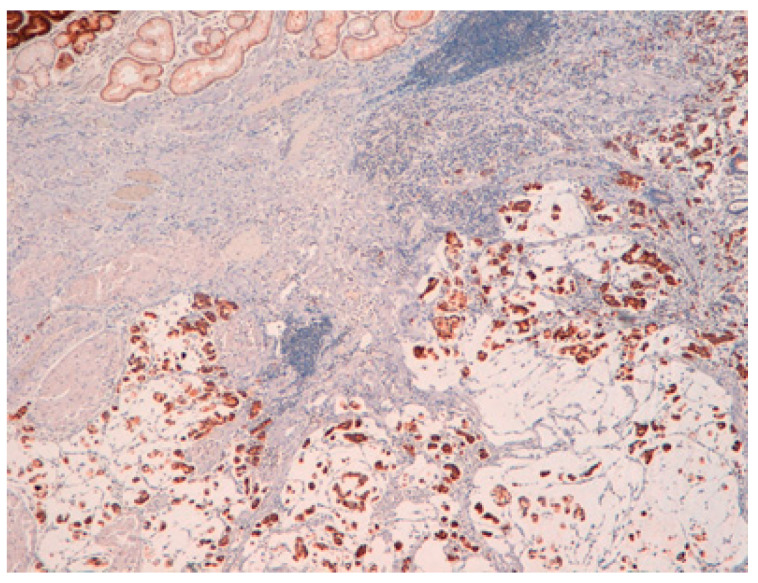
Pen A EGC showing infiltration of the submucosa layers in nodular masses.

**Figure 2 ijms-22-06652-f002:**
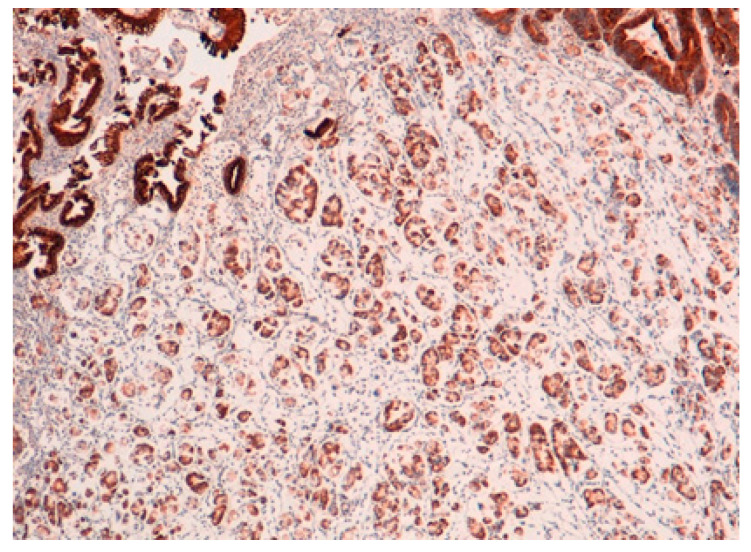
*MUC6* positivity in Pen A ECG (50×).

**Figure 3 ijms-22-06652-f003:**
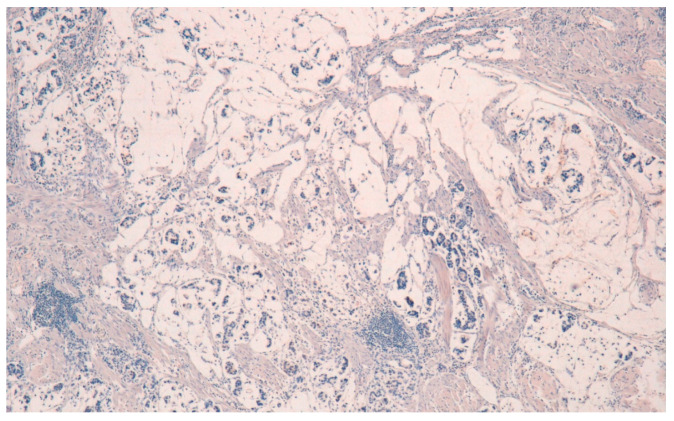
*MUC5AC* negativity in Pen A ECG (50×).

**Table 1 ijms-22-06652-t001:** Kodama’s classification.

Kodama’s Types	Description
Small mucosal	
Mucosal (M)	Intramucosal EGCs measuring less than 4 cm
Submucosal (SM)	Intramucosal EGCs minimally invading submucosa measuring less than 4 cm
Super mucosal	
Mucosal (M)	Intramucosal EGCs measuring more than 4 cm
Submucosal (SM)	Intramucosal EGCs minimally invading submucosa measuring more than 4 cm
Pen (penetrating)	
A	EGCs massively invading submucosa with nodular pattern measuring less than 4 cm
B	EGCs massively invading submucosa with saw teeth pattern measuring less than 4 cm
Mixed	Penetrating types (A or B) measuring more than 4 cm

EGC: early gastric cancer.

## Data Availability

Not applicable.

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
