# Peer review of "Molecular Alterations in Gastric Preneoplastic Lesions and Early Gastric Cancer"

_ijms, 2021, doi:10.3390/ijms22136652_

Round 1
Reviewer 1 Report
In this manuscript, Battista et al reviewed mainly on the mucins of gastric preneoplastic lesions and early gastric cancer. Below are the concerns that need to be addressed.
- The induction of CDX2 is the trigger for transdifferentiation to intestinal metaplasia. This should be presented in the intestinal metaplasia section.
- There are places where references are missing. Appropriate references should be provided.
- "Discussion" seems to be odd in a review like this. One suggestion is numbering as follows: 1. Introduction, 2. Mucins expression in normal gastric mucosa, 3. Gastric mucosal atrophy, ..., 8. Conclusions. It doesn't have to be like this, but please reconsider the structure of the manuscript.
- What do the authors mean by "informed consent was obtained from all subjects in the study"? This is a review written based on previous reports.
- There are abundant typos in the text (e.g. Oxynticand adenomas, ECG, SRG-GC, etc.). These must be corrected.
- Grammatical errors are seen throughout the text. These need to be corrected for the readers to understand the content of this review. Professional English editing service is recommended.
Author Response
- The induction of CDX2 is the trigger for transdifferentiation to intestinal metaplasia. This should be presented in the intestinal metaplasia section.
thank you very much I have done so. See 3th paragraph.
2.There are places where references are missing. Appropriate references should be provided.
Thank's a lot I have done so.
3.Thank's a lot I have done so.
4. What do the authors mean by "informed consent was obtained from all subjects in the study"? This is a review written based on previous reports. This is an error, thank's a lot, I have done so.
5. There are abundant typos in the text (e.g. Oxynticand adenomas, ECG, SRG-GC, etc.). These must be corrected. The correction the correction has been made
6. Professional English editing service has been made.
Reviewer 2 Report
· The publication addresses a clinically important issue - how to recognize gastric cancer at a very early stage and focuses on the role of mucins as markers that may influence the differentiation of subtypes of precancerous lesions with different prognoses. The role of mucins as tumor markers has been the subject of many studies which have shown that they are not specific enough to play such a role. Mucins, in various constellations are widely present on the mucous membranes of the gastrointestinal tract, respiratory tract, and urogenital tract.
· In the introduction, the authors describe the histological subtypes of gastric cancers and mention the types of molecular alterations that drive gastric cancerogenesis. At this point, they should at least mention the molecular division of GC included in The Cancer Genome Atlas
· The Authors describe the aim of the review as the search for markers that will enable early intervention in cases of serious danger of GC. The existing means of prophylaxis should be described - i.e. screening program in the countries of the highest GC occurrence and prophylactic gastrectomy in patients with CDH1 mutation (hereditary gastric cancer)
· The layout of the article is logical, although the last paragraph of the discussion (concerning early gastric cancer) is chaotic. The authors describe the cited studies one by one instead of briefly summarize their results - I suggest editing this paragraph
Author Response
- Thank's a lot for your opinion, but I do not agree in the who, 2019 the role of mucins in gastric dysplasia (and adenomas) is reiterated. And there are a lot of papers.
- Thank's a lot, I have done so (see introduction).
- Thank's a lot, I have done so.
- Thank's a lot, I have done so (see conclusion)
Round 2
Reviewer 1 Report
I think that "4. a. Gastric adenoma" should be "5. a. Gastric adenoma" ,"4. b. Gastric pyloric gland adenomas" should be "5. b. Gastric pyloric gland adenomas", and so on. Early gastric cancer will be 6., and Conclusions will be 7.
Author Response
Thank's a lot. I have done so.
Reviewer 2 Report
I'm satisfied with the reviesed version.
Author Response
Thank's a lot.